# DAGCN: Distance-based and Aspect-oriented Graph Convolutional Network for Aspect-based Sentiment Analysis

## Abstract

Aspect-based sentiment analysis (ABSA) is a task that aims to determine the sentiment polarity of aspects by identifying opinion words. Recently, numerous studies based on Graph Convolutional Networks (GCN) are used for ABSA and they mainly utilize a dependency tree to extract syntactic information. However, not all relations in a dependency tree are necessary and different context words have distinct influence on aspects. Thus, effectively utilizing syntactic information from dependency tree remains a challenging problem. In this paper, we present Distance-based and Aspect-oriented Graph Convolutional Network (DAGCN) to address the aforementioned issue, which consists of tow GCNs. Firstly, we propose a novel function called Distance-based Syntactic Value (DSV) to measure the importance of different context words syntactically and eliminate noise in the dependency tree, and thus construct Distance-based Weighted Matrix (DWM) with it. Secondly, we introduce Aspect-Fusion Attention (AF) to focus on the context words crucial for aspects in global scope and combine DWM with AF to integrate local and global syntactic information simultaneously.Finally, the first GCN (AoGCN) is designed based on the combined result to capture syntactic features and the second GCN (SaGCN) is designed with self-attention mechanism to learn semantic information. Furthermore, the Kullback-Leibler (KL) divergence loss is utilized to ensure that the features learnt by AoGCN and SaGCN are distinct. Extensive experiments on three public datasets demonstrate that DAGCN outperforms state-of-the-art models and verify the effectiveness of the proposed architecture.

## 1 Introduction

Aspect-based sentiment analysis (ABSA) is a fine-grained sentiment analysis task that aims to determine the sentiment polarity of a given aspect within a sentence. The sentiment polarity can be classified into three categories: positive, neutral, and negative. For instance, in Figure 1, the aspect *menu* can be determined to have negative sentiment polarity based on the word *limited*, while the aspect *dishes* can be classified as having positive sentiment polarity due to the word *excellent*. In fact, opinion words carry certain sentiment information and ABSA primarily focuses on identifying opinion words that are relevant to the aspect.

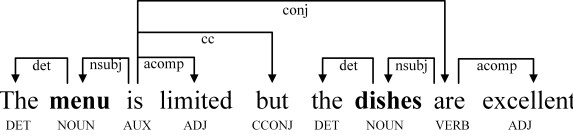

Figure 1: An example sentence with its dependency tree. There are two aspects(bolded in black) in this sentence but these aspects contain opposite sentiment polarities

Previous studies have explored heavily on attention mechanism methods and achieved promising results(Chen et al., 2017; Ma et al., 2017b; Nguyen & Le Nguyen, 2018; Liu et al., 2018; Ma et al.,

2018; Mokhosi et al., 2019). In these works, Attention mechanism is utilized to model the correlation between aspects and context words and some of context words with higher attention weights may be identified as opinion words. However, attention mechanism is sensitive to noise. High weights might be given wrongly to words that are irrelevant to the aspect, which leads to incorrect sentiment polarity predictions. For example, in Figure 1, if the combination of *excellent* and *menu* appears frequently in other sentences, attention mechanism in the current sentence might mistakenly rely on *excellent* as the evidence for predicting the sentiment polarity of *menu*. Therefore, the results of attention mechanism may contradict the ground truth.

Recently, significant efforts in ABSA have been focused on Graph Convolutional Networks (GCNs) and Graph Attention Networks (GATs), which are able to probe the dependency tree deeply (Tang et al., 2020; Zhang et al., 2022; Chen et al., 2021; Yan et al., 2021; Li et al., 2021b; Tang et al., 2022; Zhong et al., 2023). Dependency parser constructs dependency tree by analyzing the syntactic structure of sentences and establishing connections between different words from a grammatical perspective. Subsequently, GCNs and GATs aggregate node features over the adjacency matrix derived from the dependency tree to determine the sentiment polarity of the aspect. However, when using syntactic structures for this task, two issues arise: 1) Due to each element in adjacency matrix just represents whether two words are connected, GCNs can not distinguish which word is more important for aspects. 2) If only context words close to the aspect in dependency tree are considered, global syntactic information may be ignored.

In this paper, we propose a novel model named DAGCN to address the aforementioned issues. To tackle the first issue, we propose the DSV computed through Aspect-Oriented Dependency Tree(Wang et al., 2020), which creates edges between aspects and context words directly. DSV's result called Distance-based Syntactic Weight (DSW) can describe the importance of distance between aspects and context words syntactically. Then, we define Distance-based Weighted Matrix (DWM) to store DSW. The goal behind this is to learn local syntactic information by assigning higher DSW to words close to aspects in dependency tree and reduce noise by removing the edges helpless for prediction. For the second issue, considering that opinion words may exist further away from aspects, we capture global syntactic features by employing Aspect-Fusion attention (AF) to measure the importance of context words far from the aspects. Finally DWM and AF are combined to integrate local and global syntactic information and used for constructing a Graph Convolutional Network (AoGCN).

Due to the fact that GCNs over dependency tree perform poorly on the reviews with informal expression, similar to DualGCN (Li et al., 2021a), we build another GCN (SaGCN) by employing self-attention mechanism, which extracts semantic information and performs well on sentences with less structured syntax. Specifically, we incorporate a Kullback-Leibler (KL) divergence loss to ensure that the two GCNs learn distinct features, with AoGCN focusing on syntactic information and SaGCN emphasizing semantic information. Our proposed model achieves superior performance compared to state-of-the-art methods, as evidenced by experimental results on three publicly available datasets. Our contributions are summarized as follows:

- We propose DSV function to compute syntactic distance, which helps model capture local syntactic features and reduce noise in the dependency tree. We also present AF for selecting crucial words far from aspects due to its ability of learning global syntactic information. Then, we build the AoGCN over the fusion of DWM and AF, which can integrate both local and global syntactic features.

- We employ KL divergence to quantify the disparity in learned information between the two GCNs, and integrate it into the loss function to guarantee their distinct learning. Specifically, the AoGCN focuses on mining syntactic information, while the SaGCN emphasizes semantic information.

- We conduct experiments on the SemEval 2014 and Twitter datasets, and achieved state-of-the-art results, validating the effectiveness of the DAGCN architecture. To facilitate the reproducibility of our work, datasets and the source code are provided on GitHub[1].

---

[1]codes will be released when the paper is accepted.

## 2 RELATED WORK

Aspect-based sentiment analysis primarily focuses on utilizing opinion words to determine the sentiment polarity of aspects. Early works (Titov & McDonald, 2008; Thelwall & Buckley, 2013; Kim et al., 2013; Jiang et al., 2011) often relied on constructing aspect-specific sentiment lexicons or manually specified features, without incorporating syntactic features.

Recently, lots of works have focused extensively on attention mechanism to determine the semantic correlation between context words and aspects (Wang et al., 2016b; Chen et al., 2017; Ma et al., 2017b; Nguyen & Le Nguyen, 2018; Liu et al., 2018; Ma et al., 2018; Mokhosi et al., 2019; Deng et al., 2019). Ma et al. (2018) designed stacked attention mechanisms to capture both local and global features, enhancing the performance of LSTM. Furthermore, Deng et al. (2019) proposed a novel sparse self-attention mechanism to differentiate the importance of different words for sentiment polarity.

The dependency tree generated by dependency parser establishes syntactic relation between aspects and context words. Nguyen & Shirai (2015) integrated syntactic information by combining dependency relation and phrases. Wang et al. (2016a) utilized underlying syntactic information to learn a high-level feature representation. With the emergence of Graph Convolutional Networks (GCNs) and Graph Attention Networks (GATs), GCN-based and GAT-based methods have been employed to learn syntactic information from the dependency tree, and various works also considered semantic information (Zhang et al., 2019; Sun et al., 2019; Wang et al., 2020; Tang et al., 2020; Zhang et al., 2022; Chen et al., 2021; Yan et al., 2021; Li et al., 2021b; Tang et al., 2022; Zhong et al., 2023; Jiang et al., 2023). Li et al. (2021b) selected relevant knowledge from a knowledge graph and incorporated it into the dependency tree to improve its expressive power. Tang et al. (2022) considered the relationship labels of the dependency tree and proposed an adaptive fusion module for semantic information.

In addition, there have been several studies focusing on the distance between aspects and opinion words, as it is believed to contain rich syntactic and semantic knowledge (Zeng et al., 2019). In addition to utilizing semantic-relative distance to extract semantic information, Liu et al. (2022) also used the absolute distance between aspects and context words to differentiate their importance. Qi et al. (2023) computed the syntactic dependency relative distance on an undirected syntactic dependency graph.

In general, most of previous works have focused on utilizing the entire dependency tree directly, without taking into account the presence of noise and the distinct impact of different context words on aspects. By proposing a calculation of distance from the syntactic perspective and pruning the syntactic tree, DAGCN can distinguish the syntactic importance of different words and only focus on context words that have a significant impact on aspects, so that syntactic information can be used more effectively.

## 3 THE PROPOSED MODEL

Figure 2 illustrates the overview of DAGCN. Given a pair of sentence-aspect $(s, a)$, where $s = \{w_1, w_2, ..., w_n\}$ and $a = \{a_1, a_2, ..., a_m\}$ is an aspect that is part of the sentence $s$. Before feeding the input into the model, we first map each word to its embedding with the embedding table $E \in \mathbb{R}^{|V| \times d_e}$, where $|V|$ represents the size of the embedding table and $d_e$ denotes the dimension of the word embeddings. Then, an encoder such as BiLSTM or BERT is utilized to learn contextual information from the sentence. In the case of BiLSTM, the input $x$ is fed into the encoder, resulting in hidden state vectors $H = \{h_1, h_2, ..., h_n\}$, where $h_i \in \mathbb{R}^{2d_h}$ and $d_h$ represents the dimension of the hidden state vectors obtained from the unidirectional LSTM. For the BERT encoder, we construct inputs in the format required by BERT, which is "[CLS] sentence [SEP] aspect [SEP]". [CLS] and [SEP] are special tokens in BERT used for classification and sentence separation, respectively. To ensure consistency between BERT's wordpiece-based representations and syntactic dependencies based on words, we extend the dependencies of a word to all of its subwords. Next, we use $H$ as the initial node representation and input it into AoGCN and SaGCN for aggregation operations. We will now elaborate on the details of DAGCN.

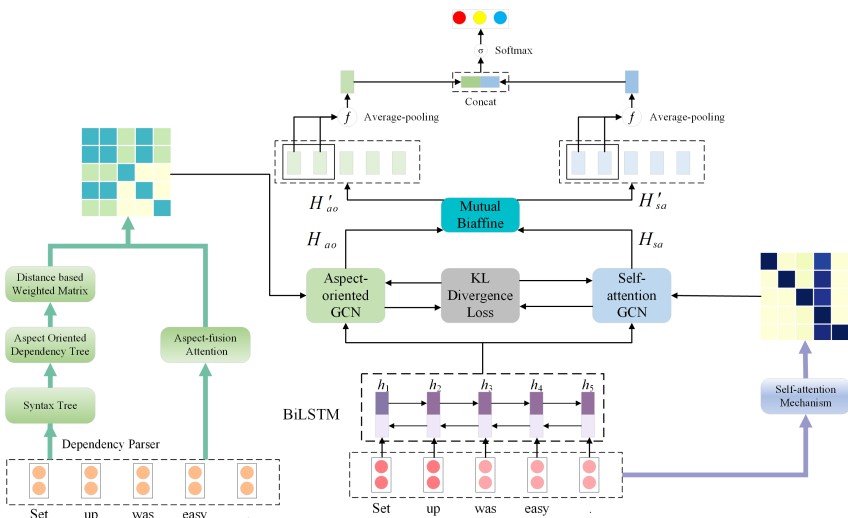

Figure 2: The overall architecture of DAGCN, which includes primarily of AoGCN and SaGCN. AoGCN is composed of Distance-based Weighted Matrix and Aspect-Fusion Attention and SaGCN is built by self-attention mechanism. Besides, the KL divergence ensures that two GCNs learn distinct information.

## 3.1 DISTANCE-BASED WEIGHTED MATRIX (DWM)

Due to the focus of ABSA lies in aspects, edges in the dependency tree that are not directly connected to the aspect are considered to provide less assistance in prediction and can be treated as noise. Inspired by R-GAT(Wang et al., 2020), the Aspect-Oriented Dependency Tree removes all edges in the dependency tree that are not directly connected to aspects, ensuring that the GCN only focuses on content related to aspects. Furthermore, given the previous studies(He et al., 2018; Zhou et al., 2021; Chen et al., 2022) that have shown that in most cases opinion words are close to the aspects in the dependency tree, we construct DWM based on the Aspect-Oriented Dependency Tree. When the context words are closer to the aspect in the dependency tree, the corresponding values in DWM are higher and vice versa.

Algorithm 1 describes the construction process of DWM. For an input sentence, we use a dependency parser to perform syntactic analysis and generate a dependency tree. Then, the construction of DWM is carried out in the following three steps. First, an empty matrix $M$ is defined to store the values between aspects and context words. Second, Distance-based Syntactic Value function (DSV) is proposed to calculate the importance of different context words for aspects, which is denoted as *DSV*. The process is as follows: if the context words are directly connected to the aspect on the dependency tree, the result of DSV $d$ is set to 1. If there is no edge directly connected to aspects in the dependency tree, we use depth-first search to find the shortest distance between the context words and the aspect, and $d$ is set to this shortest distance. Note that there may exist that the context words are not connected to aspects at all. In such cases, we set $d$ between them to infinity. Additionally, if an aspect consists of multiple words, we need to calculate $d$ between each word in an aspect and the context words separately. Third, each $d$ ranges from $[1, +\infty)$. We multiply it by a scalar $\alpha$ ($\alpha < 0$) and apply the exponential function $exp()$ to obtain Distance-based Syntactic Weight (DSW). At this stage, each DSW ranges from $[0, 1)$ and is stored in DWM. If $d$ between the context words and aspects is large, the DSW is 0, indicating that the context words are negligible for the aspect. If $d$ between the context words and aspects is small, the DSW is close to 1, indicating that the context words provide significant assistance in determining the sentiment polarity of the aspect.

There are two benefits for constructing DWM: First, it distinguishes the importance of different context words based on syntactic distance and removes irrelevant edges, which can reduce the noise. Second, words that are close to the aspect in the dependency tree are focused, which means local syntactic information is contained in it and helps identify opinion words within a local scope.

---

**Algorithm 1** Distance-based Weighted Matrix

---

**Input:** aspect $a = \{a_1, a_2, ..., a_m\}$, sentence $s = \{w_1, w_2, ..., w_n\}$, dependency Tree $T$
**Output:** Distance-based Weighted Matrix $M$
 1: construct an empty matrix $M$
 2: **for** $i = 1$ to $m$ **do**
 3:     **for** $j = 1$ to $n$ **do**
 4:        $d = DSV(a_i, w_j)$
 5:        $DSW = exp(\alpha \cdot d)$
 6:        $M_{ij} = DSW, M_{ji} = DSW$
 7:     **end for**
 8: **end for**
 9: **return** $M$

---

## 3.2 ASPECT-FUSION ATTENTION (AF)

Considering that opinion words may far from aspects and thus receive lower DSW from DWM, we propose the Aspect-Fusion Attention (AF) mechanism to distinguish more important words for aspects in the entire sentence. AF calculates attention scores by using the aspect as the query. The computation process of AF is described as follows:

$$A^{af} = avg(tanh(H_a W_{af}^a \times (K_{af} W_{af}^K)^T + b)) \tag{1}$$

Where $K$ is consistent with the output $H$ of the encoder. $W_{af}^a \in \mathbb{R}^{2d_h \times 2d_h}$ and $W_{af}^k \in \mathbb{R}^{2d_h \times 2d_h}$ are learnable weights. Note that $H_a$ is obtained from $H$ by keeping only the word embeddings at the aspect positions, i.e., $H_a = \{0, 0, .., h_{a_1}, h_{a_2}, ..., h_{a_m}, ..., 0\}$, $H_a \in \mathbb{R}^{n \times 2d_h}$. $avg()$ represents average pooling.

## 4 ASPECT-ORIENTED GCN (AoGCN)

The fusion of DWM and AF enables DWM to incorporate local and global syntactic information. The process of fusion is defined as follows:

$$M_{ij} = \begin{cases} 1, & AF_{ij} > \beta \\ M_{ij}, & otherwise \end{cases} \tag{2}$$

Where $M_{ij}$ indicates the corresponding values of DWM between $w_i$ and $w_j$. $A_{ij}^{af}$ represents the attention weight between $w_i$ and $w_j$ in $A^{af}$. $\beta$ is a hyperparameter ($\beta > 0$).

When $A^{af}$ is higher than $\beta$, it indicates that the corresponding context word is highly important for the aspect, and its distance weight in DWM needs to be increased to the maximum value of 1. Conversely, when $A^{af}$ is smaller than or equal to $\beta$, it suggests that the context word contributes less to the prediction, and its distance weight remains unchanged. Then, we multiply AF by DWM to obtain the adjacency matrix for AoGCN:

$$A^{ao} = A^{af} \times M \tag{3}$$

Where $A^{ao} \in \mathbb{R}^{n \times n}$. In $A^{ao}$, it ensures that even when the context words are far from the aspect in the dependency tree but still important for the aspect, the corresponding weights in $A^{ao}$ will be larger. Besides, if context words are close to aspects in the dependency tree and have a strong relationship with aspects, the corresponding values in $A^{ao}$ will be also larger. Specially, we treat the element of $A^{ao}$ as Comprehensive Syntactic Value (CSV), which explain the importance of context words from the perspective of overall syntax.

Based on $A^{ao}$, we can build the AoGCN. Assume that the input to the $l$-th layer is $h^{l-1}$ and the output is $h^l$, with the initial input being $h^0$. In the $l$-th layer, the hidden state $h_i^l$ of the $i$-th node can be updated by aggregating the hidden states of its neighboring nodes through the following operation:

$$h_i^l = \sigma(\sum_{j=1}^n A^{ao} W^l h_j^{l-1} + b^l) \tag{4}$$

where $W^l$ and $b^l$ are learnable weight matrix and bias, respectively. $\sigma$ is a non-linear activation function. The output of AoGCN in the last layer is denoted as $H_{ao} = \{h_1^{ao}, h_2^{ao}, ..., h_n^{ao}\}$, where $h_i^{ao}$ represents the hidden state of word $w_i$ in the last layer of AoGCN.

## 4.1 SELF-ATTENTION GCN (SAGCN)

Different from AoGCN, SaGCN utilizes self-attention mechanism to generate an attention matrix that serves as the adjacency matrix. The self-attention mechanism calculates attention scores between every pair of words in a sentence, indicating the level of correlation between two words. The calculation is shown as followed:

$$A^{sa} = \frac{QW_{sa}^Q \times (KW_{sa}^K)^T}{\sqrt{d_h}} \tag{5}$$

Where $Q$ and $K$ are the same as the input of the $l$-th layer, which is $h^{l-1}$. $W_{sa}^Q \in \mathbb{R}^{2d_h \times 2d_h}$ and $W_{sa}^K \in \mathbb{R}^{2d_h \times 2d_h}$ are learnable weight matrices. Similar to AoGCN, SaGCN ultimately obtains the graph representation $H_{sa}$.

## 4.2 BIAFFINE MODULE

To effectively interact the features learned by AoGCN and SaGCN, we employ a mutual BiAffine transformation (Tang et al., 2020) as an intermediate exchange:

$$
\begin{aligned}
H'_{ao} &= softmax(H_{ao}W_1(H_{sa})^T)H_{sa} \\
H'_{sa} &= softmax(H_{sa}W_2(H_{ao})^T)H_{ao}
\end{aligned}
\tag{6}
$$

where $W_1$ and $W_2$ are learnable parameters.

Finally, we perform average pooling and concatenation on the hidden states corresponding to aspects in the output results of the BiAffine Module, denoted as $H'_{ao}$ and $H'_{sa}$. Suppose that the aspect nodes in $H'_{ao}$ are represented by $\{h_{a_1}^{ao}, h_{a_2}^{ao}, ..., h_{a_m}^{ao}\}$ and in $H'_{sa}$ by $\{h_{a_1}^{sa}, h_{a_2}^{sa}, ..., h_{a_m}^{sa}\}$. Then, we can obtain the final representation of aspects through the following calculation:

$$h_a^{ao} = f(h_{a_1}^{ao}, h_{a_2}^{ao}, ..., h_{a_m}^{ao}) \tag{7}$$

$$h_a^{sa} = f(h_{a_1}^{sa}, h_{a_2}^{sa}, ..., h_{a_m}^{sa}) \tag{8}$$

$$h_f = [h_a^{ao}, h_a^{sa}] \tag{9}$$

Where $f(\cdot)$ represents average pooling and $[\cdot]$ denotes concatenation operation. Next, we input the final representation of aspects into a linear layer, and then the output passes through a $softmax()$ function to obtain a probability distribution vector for sentiment polarity:

$$p(a) = softmax(W_f h_f + b_f) \tag{10}$$

Where $W_f$ and $b_f$ are learnable weight matrix and bias.

## 4.3 LOSS FUNCTION

To ensure that the features learned by AoGCN and SaGCN are distinct, we introduce the KL divergence to measure the difference between them. Suppose that the probability distributions of $A^{ao}$ and $A^{sa}$ are denoted as $P(X)$ and $Q(X)$, respectively, the KL divergence loss is calculated as follows:

$$\ell_{kl}(\theta) = \sum_{x \in X} P(x)log\frac{P(x)}{Q(x)} \tag{11}$$

Where $\theta$ represents all trainable parameters. When $\ell_{kl}$ is small, it indicates that $A^{ao}$ and $A^{sa}$ have learned similar information. Conversely, when $\ell_{kl}$ is large, it suggests that $A^{ao}$ and $A^{sa}$ capture distinct syntactic and semantic features, respectively.

In addition, we also employ the standard cross-entropy loss function commonly used in ABSA, which can be defined as follows:

$$\ell_c(\theta) = - \sum_{(s,a) \in \mathcal{D}} \sum_{c \in \mathcal{C}} log p(a) \tag{12}$$

Table 1: Statistics for the three experimental datasets.

| Dataset | Split | Positive | Neutral | Negative |
|---|---|---|---|---|
| Laptop | Train | 976 | 455 | 851 |
| | Test | 337 | 167 | 128 |
| Restaurant | Train | 2164 | 637 | 807 |
| | Test | 727 | 196 | 196 |
| Twitter | Train | 1507 | 3016 | 1528 |
| | Test | 172 | 336 | 169 |

Where $\mathcal{D}$ contains all the sentence-aspect pairs and $\mathcal{C}$ is the set of sentiment polarities.

Then, we combine the KL divergence loss with the cross-entropy loss to obtain the final objective function:

$$\ell(\theta) = \ell_c(\theta) + \gamma \cdot \ell_{kl}(\theta) \tag{13}$$

Where $\gamma$ is a hyperparameter, and since we aim to capture different features from AoGCN and SaGCN, the KL divergence loss has to be sufficiently large and thus $\gamma$ must be less than 0. $\ell(\theta)$ represents the objective function, and the model parameters are optimized by minimizing the objective function.

## 5 EXPERIMENTS

### 5.1 DATASETS

We conduct experiments on three public benchmark datasets for ABSA: the Restaurant and Laptop reviews datasets from SemEval 2014 Task 4, and the Twitter dataset consisting of tweets. In the Twitter dataset, we exclude tweets with the "conflict" label. All datasets contain data with three sentiment polarities: positive, neutral, and negative. The aspect terms and sentiment polarities have been annotated in the datasets. The statistics of the three datasets are shown in Table 1.

### 5.2 IMPLEMENTATION DETAILS

We utilize Stanford's CoreNLP[2] as the dependency parser in our approach. We initialize word embeddings using 300-dimensional GloVe[3] vectors as a lookup table. Additionally, we employ 30-dimensional Part-of-Speech (POS) embeddings and 30-dimensional position embeddings, where the latter describes the distance of each word to the aspect in the sentence. The word embeddings, POS embeddings, and position embeddings are concatenated to form the input hidden states. In the encoder, the dimensions of the hidden states for BiLSTM and BERT are set to 50 and 768, respectively, with a dropout rate of 0.7. We use the bert-base-uncased[4] version of BERT. The AoGCN and SaGCN have a layer depth of 1 and a dropout rate of 0.1. We optimize the parameters using the Adam optimizer with a learning rate of 0.002. The model is trained for 50 epochs with a batch size of 16. The three hyperparameters, $\alpha$, $\beta$, and $\gamma$, are set to (-0.7, 0.9, -0.3), (-0.7, 0.3, -0.8), and (-0.2, 0.6, -0.2) respectively.

### 5.3 MAIN RESULTS

As shown in Table 2, we compared the proposed model with previous works using evaluation metrics such as accuracy and macro F1-score. These baseline models are described in detail in A.1. The experimental results demonstrate that DAGCN outperforms all baseline models on the Restaurant and Twitter datasets, and DAGCN with BERT achieves superior performance in the Laptop dataset. These results validate the effectiveness of the model architecture. In addition to comparing with some attention-based methods (i.e., IAN, RAM, and TNet) to highlight the importance of incorporating syntactic structures, we also compare with several GCN-based models. Among the

---

[2]https://stanfordnlp.github.io/CoreNLP/

[3]https://nlp.stanford.edu/projects/glove/

[4]https://github.com/huggingface/transformers

Table 2: Experimental results comparison on three publicly benchmark datasets.

| Models | Restaurant | | Laptop | | Twitter | |
|---|---|---|---|---|---|---|
| | Accuracy | Macro-F1 | Accuracy | Macro-F1 | Accuracy | Macro-F1 |
| IAN(Ma et al., 2017a) | 78.6 | - | 72.10 | - | - | - |
| RAM(Chen et al., 2017) | 80.23 | 70.80 | 74.49 | 71.35 | 69.36 | 67.30 |
| TNet(Li et al., 2018) | 80.69 | 71.27 | 76.54 | 71.75 | 74.90 | 73.60 |
| ASGCN(Zhang et al., 2019) | 80.77 | 72.02 | 75.55 | 71.05 | 72.15 | 70.40 |
| CDT(Sun et al., 2019) | 82.30 | 74.02 | 77.19 | 72.99 | 74.66 | 73.66 |
| TD-GAT(Huang & Carley, 2019) | 81.20 | - | 74.0 | - | - | - |
| BiGCN(Zhang & Qian, 2020) | 81.97 | 73.48 | 74.59 | 71.84 | 74.16 | 73.35 |
| InterGCN(Liang et al., 2020) | 82.23 | 74.01 | 77.86 | 74.32 | - | - |
| R-GAT(Wang et al., 2020) | 83.30 | 76.08 | 77.42 | 73.76 | 75.57 | 73.82 |
| DGEDT(Tang et al., 2020) | 83.90 | 75.10 | 76.80 | 72.30 | 74.80 | 73.40 |
| DualGCN(Li et al., 2021a) | 84.27 | 78.08 | 78.48 | 74.74 | 75.92 | 74.29 |
| SSEGCN(Zhang et al., 2022) | 84.72 | 77.51 | **79.43** | **76.49** | 76.51 | 75.32 |
| DAGCN | **84.72** | **78.07** | 78.96 | 75.07 | **77.10** | **75.66** |
| R-GAT+BERT(Wang et al., 2020) | 86.60 | 81.35 | 78.21 | 74.07 | 76.15 | 74.88 |
| DGEDT+BERT(Tang et al., 2020) | 86.30 | 80.00 | 79.80 | 75.60 | 77.90 | 75.40 |
| BERT4GCN(Xiao et al., 2021) | 84.75 | 77.11 | 77.49 | 73.01 | 74.73 | 73.76 |
| T-GCN+BERT(Tian et al., 2021) | 86.16 | 79.95 | 80.88 | 77.03 | 76.45 | 75.25 |
| DualGCN+BERT(Li et al., 2021a) | 87.13 | 81.16 | 81.80 | 78.10 | 77.40 | 76.02 |
| SSEGCN+BERT(Zhang et al., 2022) | 87.31 | 81.09 | 81.01 | 77.96 | 77.40 | 76.02 |
| DAGCN+BERT | **88.03** | **82.64** | **82.59** | **79.40** | **78.73** | **78.01** |

GCN-based models, our proposed model utilizes DSW and aspect-specific attention coefficients to obtain CSV, effectively distinguishing the contribution of context words in determining the aspect's sentiment polarity. Moreover, our model performs better than other GCN models that rely solely on the dependency tree structure.

Notably, when we replace the encoder of some baseline models and DAGCN with BERT, our model outperforms all of them. This improvement can be attributed to the increased information captured by word embeddings with larger dimensions. Traditional GCNs require multiple aggregations to capture features of opinion words located further from the aspect, which can lead to overfitting. In our approach, we directly connect the aspect with context words and assign CSV, enabling more targeted aggregation in the GCN, further demonstrating that our model learns more syntactic knowledge.

### 5.4 ABLATION STUDY

To validate the necessity of the proposed modules, we further conducted ablation experiments. As shown in Table 3, we first remove the KL divergence loss and only utilize the standard cross-entropy loss as the objective function. The model's performance decreases, with a reduction in accuracy of 1.58%, 3.61%, and 4.02% on the Restaurant, Laptop, and Twitter datasets, respectively. This significant drop in performance demonstrates that the KL divergence loss effectively prevents AoGCN and SaGCN from learning redundant information. Furthermore, we remove AF and the model's performance is also compromised. Without AF, the model lacks the ability to learn global syntactic information and fails to aggregate features of words distant from the aspect. Finally, by removing the DWM, we observe a significant decline in accuracy of 2.75% and 5.16% on the Restaurant and Twitter datasets, respectively. This further confirms that pruning the dependency tree allows the model to explicitly focus on the aspect, while the use of DSW differentiates the importance of context words, revealing the indispensability of local syntactic information. In summary, each proposed module contributes significantly to the overall model, and their absence leads to a performance degradation.

### 5.5 CASE STUDY

To further analysis the performance of DAGCN, we conduct a detailed analysis on real examples. As reported in Table 4, we select ATAELSTM, IAN, and ASGCN to compare their classification

Table 3: Experimental results of ablation study.

| Models | Restaurant | | Laptop | | Twitter | |
|---|---|---|---|---|---|---|
| | Accuracy | Macro-F1 | Accuracy | Macro-F1 | Accuracy | Macro-F1 |
| DAGCN | **84.72** | **78.07** | **78.96** | **75.07** | **77.10** | **75.66** |
| w/o KL divergence loss | 83.38 | 75.53 | 76.11 | 71.95 | 74.00 | 72.61 |
| w/o aspect-fusion attention | 82.84 | 75.19 | 76.58 | 72.27 | 74.00 | 72.85 |
| w/o distance based weighted matrix | 82.39 | 73.85 | 78.16 | 74.27 | 73.12 | 70.61 |

Table 4: Case studies of our DAGCN model compared with state-of-the-art baselines.

| Sentence | ATAE-LSTM | IAN | ASGCN | DAGCN | Target |
|---|---|---|---|---|---|
| it's fast, light, and simple to *use*. | **P** | **P** | **P** | **P** | P |
| our *waiter* was friendly and it is a shame that he didn't have a supportive staff to work with. | N | **P** | **P** | **P** | P |
| I am still in the process of learning about its *features*. | P | P | **O** | **O** | O |
| Did not enjoy the new windows 8 and *touchscreen functions*. | P | P | P | **N** | N |
| When the *dish* arrived it was blazing with *green chillis*, definitely not edible by a human. | P, O | P, P | P, P | **N, N** | N, N |

capabilities with DAGCN. In each example, the aspect is indicated in italics, and the notations P, N, and O represent positive, negative, and neutral sentiment, respectively.In the first example, the aspect is *"use"* and its corresponding opinion words are *"fast"*, *"light"*, and *"simple"*. These words have similar positive sentiment, making it unambiguous for attention-based methods to quickly distinguish the sentiment polarity. In the second example, the sentiment polarity of *"friendly"* and *"shame"* is opposite, which may confuse attention-based methods and lead to wrong classification, as they might mistakenly focus on *"shame"* as the judgement basis. In the fourth example, attention-based methods fail to capture the impact of the negative word *"not"* on sentiment polarity, while ASGCN also overlooks the syntactic relationship between the aspect and opinion words. However, DAGCN assigns a higher DSW to *"not enjoy"*, enabling the model to make accurate predictions, highlighting the importance of local syntactic information in ABSA. Similarly, in the last example, the two aspects are separated by punctuation from *"not edible"*, resulting in a greater syntactic distance between them. DAGCN, utilizing aspect-fusion attention, comprehensively searches for opinion words from a global perspective and successfully make accurate prediction. Additionally, DAGCN demonstrates strong discriminative power for sentences with neutral sentiment in example 3.

## 6 CONCLUSION

In this paper, we have prensted a novel DAGCN model. By defining Distance-based Syntactic Value function (DSV), Distance-based Syntactic Weight (DSW) is able to make context words close to the aspects have higher impact, and it could also eliminate noise in the dependency tree. Based on it, we construct Distance-based Weighted Matrix (DWM) to store DSWs and update DWM by Aspect-Fusion Attention (AF) to take context words far from aspects into account. Then DWM and AF are combined for building AoGCN, which can incorporate both local and global syntactic information. Through their combined use, AoGCN obtains Comprehensive Syntactic Value (CSV) that facilitates the focus on opinion words highly correlated to the aspects. Besides, inspired by previous work, we define a SaGCN with self-attention mechanism to deal with some reviews with unstructured syntax. Furthermore, KL divergence is integrated into the loss function to guarantee distinct learning for AoGCN and SaGCN. Compared with other baseline models, DAGCN achieves superior performance on public datasets, which demonstrates the effectiveness of the proposed architecture.

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

# A  APPENDIX

## A.1  BASELINE MODELS

To thoroughly evaluate the effectiveness of our proposed model, we compare DAGCN against state-of-the-art baselines, including:

1. **IAN** (Ma et al., 2017a) proposes a new approach for ABSA by separately modeling the targets and contexts using interactive attention networks.

2. **RAM** (Chen et al., 2017) integrates a recurrent neural network with a weighted-memory mechanism to capture sentiment features.

3. **TNet** (Li et al., 2018) combines a BiLSTM layer with a CNN layer to extract salient features from transformed word representations.

4. **ASGCN** (Zhang et al., 2019) first employs a GCN to learn aspect representations in aspect based sentiment analysis task.

5. **CDT** (Sun et al., 2019) uses a Bi-LSTM for learning sentence features and a GCN is applied to the dependency tree to enhance the embeddings.

6. **TD-GAT** (Huang & Carley, 2019) utilizes the dependency relationship among words to propagate sentiment features directly from the syntactic context of an aspect target.

7. **BiGCN** (Zhang & Qian, 2020) proposes a bi-level interactive graph convolution network to covolutes over hierarchical syntactic and lexical graphs.

8. **InterGCN** (Liang et al., 2020) constructs a heterogeneous graph for each instance by leveraging aspect-focused and inter-aspect contextual dependencies

9. **R-GAT** (Wang et al., 2020) encodes syntax information through a aspect-oriented dependency tree structure and introduces dependency relation into convolution.

10. **DGEDT** (Tang et al., 2020) proposes a dependency graph enhanced dual-transformer network that utilizes a dual-transformer structure to mutually reinforce the flat and graph-based representations.

11. **DualGCN** (Li et al., 2021a) utilizes two GCNs to learn syntactic information and semantic information, respectively.

12. **SSEGCN** (Zhang et al., 2022) proposes an aspect-aware attention mechanism with self-attention to learn aspect- related and global semantics of a sentence and then combines them with syntactic information.

13. **R-GAT+BERT** (Wang et al., 2020) is the R-GAT model whose encoder is replaced by a pre-trained BERT.

14. **DGEDT+BERT** (Tang et al., 2020) is the DGEDT model whose encoder is replaced by a pre-trained BERT.

15. **BERT4GCN** (Xiao et al., 2021) integrates the contextual features output from BERT and the syntactic knowledge from dependency graphs.

16. **T-GCN+BERT** (Tian et al., 2021) utilizes attention and layer ensemble to explicitly consider dependency types in the graph.

17. **DualGCN+BERT** (Li et al., 2021a) is the DualGCN model whose encoder is replaced by a pre-trained BERT.

18. **SSEGCN+BERT** (Zhang et al., 2022) is the SSEGCN model whose encoder is replaced by a pre-trained BERT.

## A.2  VISUALIZATION

In Figure 3, we visualize two examples to investigate whether DAGCN can assign higher weights to opinion words. In the first example, we observe that the aspect *"battery"* is in close proximity to the opinion word *"longer"* in the dependency tree. The DWM assigns a higher DSW to this word, and the aspect-fusion attention helps the DWM distinguish which word is most important within the local context, ultimately focusing on *"longer"*. In cases where longer texts contain a larger number

of words, the model may struggle to identify opinion words clearly, especially when the distance between opinion words and the aspect in dependency tree is large. To address this, we visualize a sentence from a longer text where the aspect is *"30" hd monitor"*. In this sentence, both the word *"fresh"* and the opinion words (*"helps"*) could be considered informative for classification. However, by leveraging the DWM's ability to extract local syntactic information, our model alleviates confusion and improves prediction accuracy.

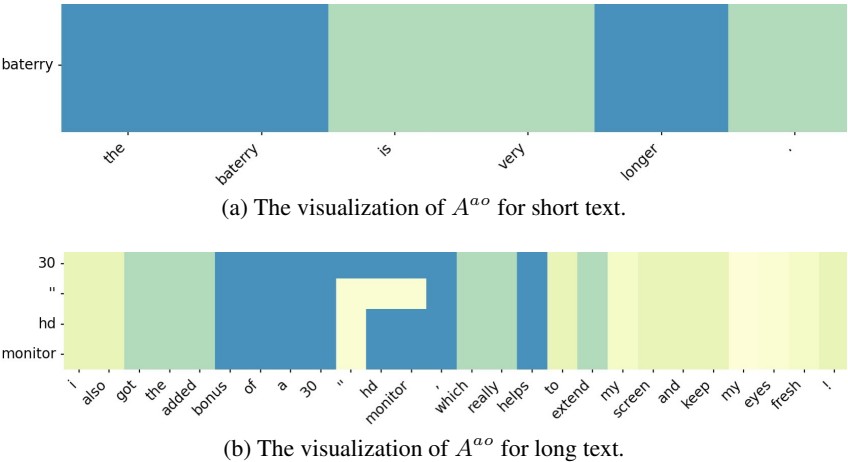

(a) The visualization of $A^{ao}$ for short text.

(b) The visualization of $A^{ao}$ for long text.

Figure 3: Two visualization of learnt comprehensive syntactic distance weights for short text and long text.

### A.3 EFFECT OF THE DAGCN LAYER NUMBER

In this section, we investigate the impact of DAGCN layer number on the performance. Figure 4 illustrates the changes in accuracy and macro-F1 scores on the Restaurant and Laptop datasets as the layer number varies from 1 to 5. From the results, we can observe that the model performs optimally when the number of GCN layers is 1. As the number of layers increases, particularly when it reaches 5, the performance diminishes. This is attributed to the direct connection between the aspects and context words through the construction of DWM. With fewer layers, the model avoids excessive aggregation operations, whereas a higher number of layers can lead to overfitting.

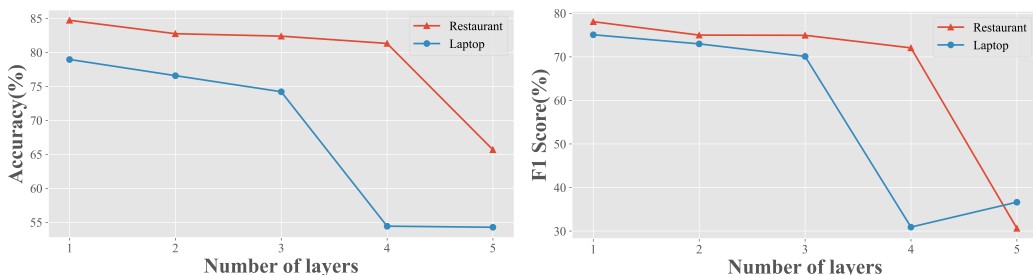

Figure 4: Effect of the number of DAGCN layers.

