# OpenReview forum: "DAGCN: Distance-based and Aspect-oriented Graph Convolutional Network for Aspect-based Sentiment Analysis"
_ICLR.cc/2024/Conference — ICLR 2024 Conference Withdrawn Submission_

### Official Review · Reviewer_QpvE · 2023-10-28

**Soundness:** 2 fair
**Presentation:** 2 fair
**Contribution:** 1 poor
**Rating:** 1
**Confidence:** 5

**Summary:**

The paper introduces a novel model called DAGCN that aims to enhance aspect-based sentiment analysis (ABSA) by effectively utilizing syntactic information from dependency trees. The document discusses the limitations of previous approaches and presents DAGCN as a solution. The model consists of two Graph Convolutional Networks (GCNs) - Aspect-oriented GCN (AoGCN) and Self-Attention GCN (SaGCN). AoGCN captures syntactic features by constructing a Distance-based Weighted Matrix (DWM) using Distance-based Syntactic Value (DSV) to measure the importance of context words, while SaGCN focuses on learning semantic information. DAGCN combines the DWM with Aspect-Fusion Attention (AF) to integrate local and global syntactic information. The two GCNs are designed to learn distinct features using the Kullback-Leibler (KL) divergence loss. Experimental results on three public datasets demonstrate that DAGCN outperforms baseline models in ABSA.

**Strengths:**

The paper presents a detailed elucidation of the proposed model, complete with implementation specifics and evaluation results. It underscores the efficacy of incorporating syntactic information for aspect-based sentiment analysis.

**Weaknesses:**

1. The most glaring concern with this paper is its pronounced temporal detachment from current trends. The idea of leveraging syntax dependency features to assist in modeling long-range dependencies, thereby facilitating granular sentiment analysis, has already been extensively probed within the NLP community. This approach was well-addressed and effectively deployed in various NLP tasks such as Relation Extraction nearly 5-6 years ago.

2. The angle this paper pursues, namely that "not all relations in a dependency tree are essential and different context words exert varied influences on aspects", has been amply investigated by predecessors. Regrettably, the authors have not sufficiently covered this breadth of prior exploration in their literature survey.

3. The methodology put forth lacks substantial novelty. The techniques exhibited are fundamentally simple amalgamations of those proposed half a decade ago.

4. The experimental section of the work is somewhat underwhelming. The authors have not delved deep into analysis and exploration. Specifically, there's an evident gap in addressing how their proposed model tackles the challenge of discerning necessary relations in a dependency tree and understanding the diverse influence of context words on aspects.

**Questions:**

1. In reference to the weaknesses highlighted above, how do the authors reconcile the evident temporal gap between their work and prior seminal research?

2. Given that many of the explored themes have seen extensive examination in prior works, can the authors provide a more comprehensive survey detailing how their approach either complements or surpasses these earlier endeavors?

3. Can the authors elucidate on their choice of methodology, specifically regarding its novelty, in the context of existing solutions introduced years prior?

4. It would be beneficial for the readers if the authors could furnish deeper insights into how their model specifically addresses the challenges of discerning essential relations and gauging the varied influence of context words on aspects.

---

### Official Review · Reviewer_U2MN · 2023-10-28

**Soundness:** 2 fair
**Presentation:** 2 fair
**Contribution:** 1 poor
**Rating:** 3
**Confidence:** 4

**Summary:**

The main points of this paper revolve around the proposal of the DAGCN model for aspect-based sentiment analysis (ABSA).
The authors highlight the limitations of previous studies that utilize Graph Convolutional Networks (GCNs) and attention mechanisms in extracting syntactic information from dependency trees.
To address these limitations, the authors propose the DAGCN model, which consists of two GCNs: Aspect-Oriented GCN (AoGCN) and Self-Attention GCN (SaGCN).
Experimental results on three datasets demonstrate that DAGCN outperforms existing models in ABSA.

**Strengths:**

1. The paper introduces a novel model called DAGCN for aspect-based sentiment analysis.
2. The experimental results show that DAGCN outperforms existing models in ABSA.

**Weaknesses:**

1. The motivation is unconvincing as several issues within the integration of dependency trees for aspect-based sentiment analysis have been fully explored:

	a. The claim that the "attention mechanism is sensitive to noise" lacks persuasiveness. It remains questionable how frequently occurring combinations in other sentences could significantly impact the attention mechanism within the current sentence. Furthermore, the absence of concrete examples or references weakens this argument.

	b. The assertion that existing GCN-based methods cannot distinguish the importance of words for aspects is unconvincing. Notably, the GAT-based method [1,2,3] has demonstrated the capability to assess the necessity and influence of relations within a dependency tree on aspects.

	c. There is also a lack of acknowledgment of several works [4,5,6] that have dedicated significant attention to leveraging global information to enhance aspect-based sentiment inference.

2. It is paramount to question whether dependency trees are indeed the linchpin of aspect-based sentiment analysis, given that there exist various other methods that can significantly enhance task performance, such as the utilization of knowledge graphs [10] or generative models [9].

3. Some pivotal and recent related works, such as [2,3,5,6], have been overlooked.

4. The proposed method lacks novelty, as it essentially combines existing mechanisms, such as attention fusion [7,8], and aspect-oriented GCN [1].

5. The experiments fall short of providing an in-depth analysis of how the proposed method addresses the challenges of distinguishing necessary relations within a dependency tree and understanding the diverse influence of context words on aspects.

[1] Relational Graph Attention Network for Aspect-based Sentiment Analysis

[2] Learn from Syntax: Improving Pair-wise Aspect and Opinion Terms Extraction with Rich Syntactic Knowledge

[3] Graph convolutional network with multiple weight mechanisms for aspect-based sentiment analysis

[4] SSEGCN: Syntactic and semantic enhanced graph convolutional network for aspect-based sentiment analysis

[5] Convolution over hierarchical syntactic and lexical graphs for aspect-level sentiment analysis

[6] Global inference with explicit syntactic and discourse structures for dialogue-level relation extraction

[7] Effective attention modeling for aspect-level sentiment classification

[8] Content attention model for aspect-based sentiment analysis

[9] A generative language model for few-shot aspect-based sentiment analysis

[10] Knowledge graph augmented network towards multiview representation learning for aspect-based sentiment analysis

**Questions:**

see weakness.

---

### Official Review · Reviewer_dd5U · 2023-10-31

**Soundness:** 2 fair
**Presentation:** 3 good
**Contribution:** 2 fair
**Rating:** 5
**Confidence:** 3

**Summary:**

The paper proposes a model called DAGCN for aspect-based sentiment analysis (ABSA) using Graph Convolutional Networks (GCNs). The model addresses the challenge of effectively utilizing syntactic information from dependency trees by introducing a distance-based weight matrix (DWM).

In DWM, 1 means aspect word and context word are  in very close distance. 0 means aspect word and context word are in very far distance.

**Strengths:**

- The paper is well-written and easy to follow.
- The paper propose a new approach to get adjacency matrix of GCN.
- The authors claim they will release the code if paper is accepted.

**Weaknesses:**

- I think this paper has some innovation, but it is not innovative enough. Because it is common to use GCN and attention mechanisms for sentiment analysis. Compared with previous work, the contribution of this paper is only to redefine the adjacency matrix of GCN.
- The claim of SOTA  on Twitter datasets is not convincing. Because some baselines are not considered, such as [1].




[1] Ma F, Hu X, Liu A, et al. AMR-based Network for Aspect-based Sentiment Analysis[C]//Proceedings of the 61st Annual Meeting of the Association for Computational Linguistics (Volume 1: Long Papers). 2023: 322-337.

**Questions:**

- There are many latest SemEval datasets, why use a very old dataset SemEval 2014?
- Could you give an example of "noise" mentioned in your paper?

---

### Official Review · Reviewer_kTiw · 2023-10-31

**Soundness:** 2 fair
**Presentation:** 3 good
**Contribution:** 2 fair
**Rating:** 3
**Confidence:** 3

**Summary:**

This paper presents a distance-based and aspect-oriented graph convolutional network for aspect-based sentiment analysis. The authors conduct extensive experiments on four benchmark datasets. The results show the effectiveness of the proposed method. The target problem is interesting and well to be investigated.

**Strengths:**

1. The target problem is interesting and well to be investigated.
2. The results show the effectiveness of the proposed method.

**Weaknesses:**

1. The novelty is limited. For example, using the GCN for aspect-based sentiment analysis has been widely investigated. The proposed method is an incremental contribution compared to Zhong et al. (2023) and Jiang et al. (2023).
2. In Table 2, some latest works should be compared and discussed.
Zhong et al., 2023. Knowledge Graph Augmented Network Towards Multiview Representation Learning for Aspect-Based Sentiment Analysis.
Yu et al., 2023. A novel weight-oriented graph convolutional network for aspect-based sentiment analysis.
Ma et al., 2023. AMR-based Network for Aspect-based Sentiment Analysis.
Wu et al., 2023. Improving aspect-based sentiment analysis with Knowledge-aware Dependency Graph Network.

**Questions:**

1. The novelty is limited. For example, using the GCN for aspect-based sentiment analysis has been widely investigated. The proposed method is an incremental contribution compared to Zhong et al. (2023) and Jiang et al. (2023).
2. In Table 2, some latest works should be compared and discussed.
Zhong et al., 2023. Knowledge Graph Augmented Network Towards Multiview Representation Learning for Aspect-Based Sentiment Analysis.
Yu et al., 2023. A novel weight-oriented graph convolutional network for aspect-based sentiment analysis.
Ma et al., 2023. AMR-based Network for Aspect-based Sentiment Analysis.
Wu et al., 2023. Improving aspect-based sentiment analysis with Knowledge-aware Dependency Graph Network.